# Time, momentum, and energy resolved pump-probe tunneling spectroscopy of two-dimensional electron systems

H. M. Yoo[1], M. Korkusinski[2], D. Miravet [3], K. W. Baldwin[4], K. West[4], L. Pfeiffer[4], P. Hawrylak[3] & R. C. Ashoori [1] ✉

Real-time probing of electrons can uncover intricate relaxation mechanisms and many-body interactions in strongly correlated materials. Here, we introduce time, momentum, and energy resolved pump-probe tunneling spectroscopy (Tr-MERTS). The method allows the injection of electrons at a particular energy and observation of their subsequent decay in energy-momentum space. Using Tr-MERTS, we visualize electronic decay processes, with lifetimes from tens of nanoseconds to tens of microseconds, in Landau levels formed in a GaAs quantum well. Although most observed features agree with simple energy-relaxation, we discovered a splitting in the nonequilibrium energy spectrum in the vicinity of a ferromagnetic state. An exact diagonalization study suggests that the splitting arises from a maximally spin-polarized state with higher energy than a conventional equilibrium skyrmion. Furthermore, we observe time-dependent relaxation of the splitting, which we attribute to single-flipped spins forming skyrmions. These results establish Tr-MERTS as a powerful tool for studying the properties of a 2DES beyond equilibrium.

Nonequilibrium response in a material driven by laser and electrical pulses can reveal many-body effects involving strong interactions between electrons and their surrounding medium. In superconductors, the relaxation dynamics of excited quasiparticles reflects the strength of electron interactions with phonons or other collective excitations, such as spin fluctuations, that may give rise to unconventional superconductivity[1–4]. The measurement of relaxation times in other correlated states also permits investigation of their electronic origins[5–7]. Furthermore, pumping a system can tune electron-lattice interactions[8,9] and magnetic couplings[10,11], providing a new pathway for exploring many-body correlations that are undetectable in an equilibrium state. While prior optical pump-probe studies[3,12] have demonstrated the capability of probing dynamics occurring in correlated materials on a picosecond to subpicosecond timescale, there are limitations in applying this laser-based technique to a two-dimensional (2D) electronic system. First, a high-intensity laser easily heats up the

sample and prevents the study of temporal dynamics at millikelvin temperatures, where a variety of delicate correlated phases emerge. In addition, the laser pulse excites carriers into any available states that have transition energy equal to the photon energy, making it difficult to pump carriers only into a specific state. An alternative method, pump-probe scanning tunneling microscopy[13], can be utilized for investigating the nonequilibrium properties in 2D materials. However, this method is sensitive to the surface of a sample and is not easily applicable to high-quality 2D materials that are often encapsulated with insulating dielectrics. Moreover, for 2D conductors, local pumping from a scanning tip would lead to rapid lateral charge spreading, making it difficult to study dynamics that do not involve transport.

Here, we present a new method — time, momentum, and energy resolved pump-probe tunneling spectroscopy (Tr-MERTS) — that uses planar tunneling and allows visualization of nonequilibrium states in a two-dimensional electronic system (2DES). Tr-MERTS employs short

[1]Department of Physics, Massachusetts Institute of Technology, Cambridge, MA 02139, USA. [2]Emerging Technologies Division, National Research Council of Canada, Ottawa, ON K1A 0R6, Canada. [3]Department of Physics, University of Ottawa, Ottawa, ON K1N 6N5, Canada. [4]Department of Electrical Engineering, Princeton University, Princeton, NJ 08544, USA. ✉e-mail: ashoori@mit.edu

duty cycle pulses and functions in the millikelvin temperature range. In addition, the fine control of electrical pulses utilized in Tr-MERTS permits submillivolt energy resolution[14], temporal resolution of $\Delta t \lesssim 100$ ns, and precise tunability of pumping electron densities. Finally, since the pumping energy can be tuned by the height of an applied pulse, electrons can be pumped into a specific energy state even for a system with equidistant energy levels. In this study, we demonstrate the capability of Tr-MERTS via its application to interacting electrons in Landau levels (LLs) formed in a GaAs 2DES. We employed a GaAs 2DES due to its high-mobility and ease of sample fabrication, making it an ideal prototypical 2DES for demonstrating our pump-probe method. Using Tr-MERTS, we explore the spin-dependent temporal dynamics in a LL. Surprisingly, we uncover a transient level splitting that arises from the formation of a locally excited electronic droplet triggered by the addition of a minority-spin into a ferromagnetic ground-state.

## Results

To perform Tr-MERTS measurements, we apply a sequence of three pulses to a bilayer tunnel device (Fig. 1a, b). The first "pump" pulse ejects electrons from one of the two layers (source) and pumps these electrons into the other layer (target). We set the energy of the electrons injected into the target during pumping by varying the height of a pump pulse (see Fig. 1c). Immediately after the pump pulse, we apply

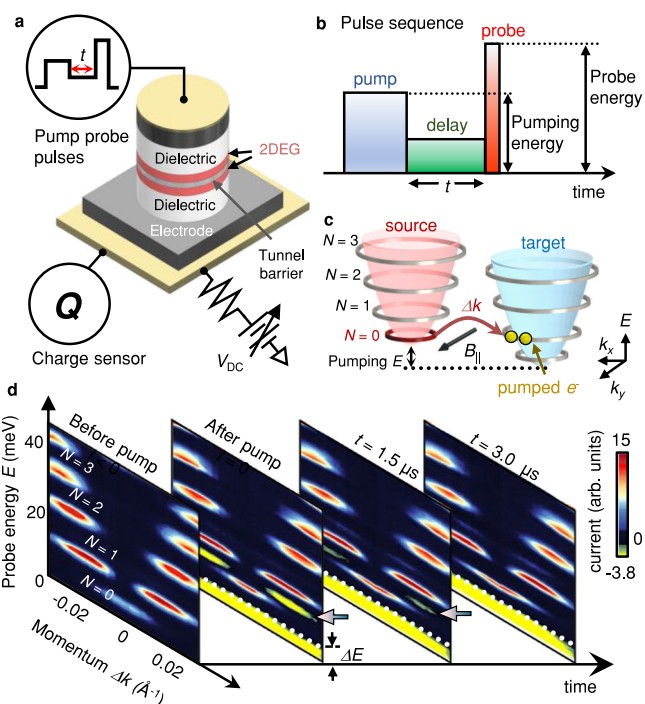

**Fig. 1 | Tr-MERTS setup and spectra. a** The tunnel device consists of two 2DESs separated by a thin tunnel barrier (see Methods). We apply a sequence of electrical pulses across the device and detect the image charge of the tunneled electrons[14–16]. **b** Schematic representation of the pulse sequence. The first "pump" pulse ejects electrons from the source and pumps these electrons into the target. The second "delay" pulse turns off tunneling between the source and target during the time delay ($t$). The last "probe" pulse measures the current ($I$) flowing into the target over a short period of time[14]. **c** A cartoon depicting a pumping process. During the pump pulse, electrons in the $N = 0$ LL in the source are pumped into a higher LL in the target. **d** A sequence of Tr-MERTS spectra measured as a function of $t$ before and after pumping the $N = 1$ LL at $B_\perp = 5.1$ T and $T = 50$ mK. Arrows indicate transient tunneling current. $\Delta E$ arises from a charging effect that is proportional to the density of pumped electrons (see Methods). Yellow features below $\Delta E$ are not transient and result from tunneling between the $N = 0$ LLs in the source and target (See Supplementary Fig. 5 for a detailed explanation).

a second "delay" pulse to induce an out-of-resonance condition, in which the pumped states in the target and the available states in the source are misaligned in energy. In such a condition, the pumped electrons cannot flow back to the source and remain in the target (see Supplementary Fig. 4 for details). After the delay pulse, we apply a short "probe" pulse and measure the tunneling current ($I$) by detecting an initial rise in the image charge of tunneled electrons[14–16]. We use precisely defined pulses with 1–99% rise time of ~4 ns, using specialized home-built electronics that eliminates all ringing. Unlike optical pump-probe experiments, the applied pulses in Tr-MERTS do not overlap in time, and we thus measure current solely driven by the probe pulse (see Supplementary Fig. 4). Finally, by controlling the time delay ($t$) between the pump and probe pulses, we monitor time-dependent changes in tunneling spectra. We simultaneously acquire momentum space information by applying an in-plane magnetic field $B_\parallel$ that shifts the momentum $\Delta k = eB_\parallel d/\hbar$ of the tunneling electrons (Fig. 1c), where $d$ is the physical separation between the target and the source layers[16].

Figure 1d shows a sequence of Tr-MERTS spectra measured before ($t < 0$) and after ($t \geq 0$) pumping the target in an applied perpendicular magnetic field $B_\perp = 5.1$ T and at a temperature $T = 50$ mK. The Landau level filling factors of the source and target are $\nu = 0.35$ and $\nu = 1$ respectively, and the electrons injected from the source are nearly fully spin-polarized[17,18]. At $t < 0$, the spectra show quantized LLs in energy and momentum space[16]. When we drive the target out of equilibrium by pumping electrons into the $N = 1$ LL, we observe an upward energy shift $\Delta E$ in the spectrum. This $\Delta E$ arises from a charging energy, proportional to the density of the pumped electrons $n^{pump}$ (see Methods). Furthermore, the spectrum shows a transient negative current at energy below the $N = 1$ LL (see yellow features in Fig. 1d). This transient current arises from the pumped electrons flowing back to the source under a specific condition: When the pumped energy level in the target matches available energy states in the source, the pumped electrons tunnel back to the source (see Supplementary Fig. 5 for a detailed explanation). The $\Delta k$ distribution of the transient current reveals the information about pumped energy states (i.e. the $N = 1$ LL), and its intensity is proportional to the number of the excited electrons. At large $t$, the transient current disappears as the pumped electrons decay to the $N = 0$ LL in the target.

To study the relaxation dynamics of electrons between the two lowest ($N = 0$ and 1) LLs, we measured the target filling factor ($\nu$) dependence of Tr-MERTS spectra. Fig. 2a shows TR-MERTS spectrum measured as functions of $E$, $\Delta k$, and $\nu$ immediately after pumping the $N = 1$ LL. Each surface in the figure displays a constant $E$, $\Delta k$, or $\nu$ cut, and the surface color represents the intensity map of the tunneling current. In a constant momentum cut taken at $\Delta k = 0.014$ Å$^{-1}$, we observe a transient negative current at energy slightly below the $N = 1$ LL at $\nu = 1$ and $\nu \geq 1.5$. At other filling factors in the target, these transient features are absent because electrons can relax between the LLs at a rate faster than the tunneling rate. For more quantitative analysis, we measured the $t$ dependence of the spectrum for different values of $\nu$ (see Fig. 2b–d). At $\nu = 1$ and 5/3, the transient current persists up to $t \approx 1$ μs and 10 μs, respectively. On the other hand, at $\nu = 3/2$, the transient current disappears at $t \approx 0.1$ μs. This behavior suggests that the relaxation time is substantially slowed down at integer and fractional fillings, where the ground-state properties change significantly owing to strong electronic interactions[17,18].

Given that the spectral weight of the transient current is proportional to the density of excited electrons, we deduce the decay constant ($\tau$) by fitting an exponential function $e^{t/\tau}$ to the time dependence of the energy integrated transient current (see Supplementary Fig. 6). Fig. 2e shows $\tau$ for $1 \leq \nu < 2$. A simple model based on Fermi's golden rule for a two-level system predicts monotonically increasing $\tau$ because the number of available states ($\rho_f$) in the lower LL linearly decreases with increasing $\nu$. However, we observe more than an order of magnitude reduction in $\tau$ at $\nu \approx 1.2$ and 1.5. Such behavior suggests that a spin-flip

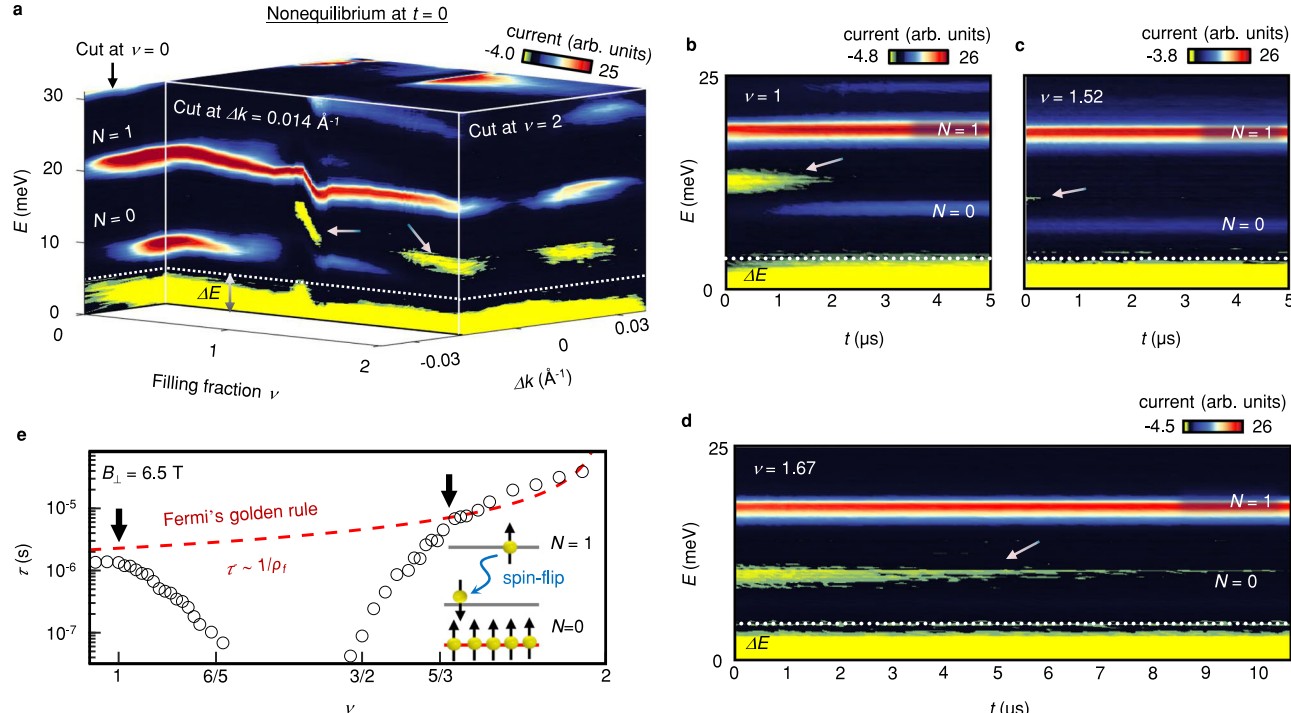

**Fig. 2 | ν dependence of Tr-MERTS spectra. a** 4D visualization of Tr-MERTS spectrum measured as functions of $E$, $\Delta k$, and $ν$. The spectrum is taken immediately after pumping the $N=1$ LL at $B_\perp = 6.5$ T and $T = 50$ mK. The vertical axis is $E$, proportional to the height of the probe pulse. Horizontal and depth axes are $ν$ of the target and $\Delta k$, respectively. Each surface shows a constant $\Delta k$, $ν$, or $E$ cut. Colors represent the intensity of the tunneling current. The $ν$ of the source is fixed at 0.35. **b–d** Tr-MERTS spectra measured as functions of $E$ and $t$ at $\Delta k = 0.014$ Å$^{-1}$. **e** $ν$ dependence of relaxation time $τ$. The red dashed line is a fit based on Fermi's golden rule for decays, inversely proportional to the density of states ($ρ_f$) available in the lower energy level. The deviation from the fit grows at $ν>1$ and $ν<5/3$ (see black arrows) due to a sharp change in the ground-state spin polarization[17,18]. The inset illustrates a slow relaxation process involving spin-flips. Red and grey horizontal lines represent the occupied and unoccupied states in the $N=0$ LL, respectively.

process is responsible for the slow relaxation time observed around $ν=1$ and $ν\geq5/3$. In these ranges of $ν$, electrons in the occupied states form nearly fully spin-polarized states[17,18], with unoccupied states in the $N=0$ LL spin-polarized in the opposite direction. In such a case, the spin of pumped electrons is antiparallel to that of unoccupied states in the $N=0$ LL in the target because the electrons are injected from a nearly spin-polarized source[17]. As a result, relaxation requires a slow scattering process involving the emission of phonons that can mediate the spin-flips via the spin-orbit interaction[19,20]. Between $ν=1.2$ and 1.5, the unoccupied states in the $N=0$ LL are unpolarized. Therefore, $τ$ decreases rapidly in these trivial (unpolarized) states because the relaxation process does not require spin-flips.

To corroborate this assertion, we measured the $T$ dependence of the relaxation at $ν=1$ because the magnetization of a 2DES is sensitive to temperature changes[21]. Tr-MERTS spectra in Supplementary Fig. 7a show a faster decay of transient current (see yellow features) with increasing $T$. This trend is consistent with the spin-dependent relaxation process described above. As $T$ increases, thermally excited magnons depolarize the 2DES at $ν=1$[22]. Likewise, the unoccupied states (holes) in the $N=0$ LL become spin-depolarized. In this case, the decay no longer requires a spin-flip process and becomes faster. The $T$ dependence at $ν=5/3$ also displays a similar behavior (see Supplementary Fig. 7b for details).

To investigate the nonequilibrium dynamics in more detail, we vary the width of the pump pulse from 0 to -1 μs and fine-tune the $n^{pump}$ injected into the target. In the Tr-MERTS spectra measured immediately after pumping the $N=1$ LL in the target at $ν=1$, we observe an unexpected double-peak structure, with a higher energy peak emerging at higher pumping density in the $N=1$ LL at $n^{pump} \sim 4.6 \times 10^9$ cm$^{-2}$ (see the dashed circle in Fig. 3a). A constant wavevector cut of the spectrum measured as a function of $n^{pump}$ in Fig. 3b clearly shows a

crossover between the two anomalous peaks in the $N=1$ LL at intermediate $n^{pump}$. Furthermore, we have detected these two split peaks in various $B_\perp$, but they are absent away from $ν=1$ (see Supplementary Fig. 8). Lastly, the energy scaling of the splitting between the two peaks (see Fig. 3c) suggests that it originates from electronic interactions in the $ν=1$ state.

The formation of other equilibrium states such as skyrmions[17,21,23] cannot be responsible for the splitting because the tunneling spectrum measured without pumping shows no signature of the splitting when the static electron density $n^{static}$ is tuned away from $ν=1$ (see Fig. 3d and Supplementary Fig. 9). In addition, after a sufficiently long-time delay $t=300$-500 ns, the split peaks disappear and transform back to a single energy peak (see Fig. 3e). This temporal behavior suggests that the splitting is characteristic to the system driven out of equilibrium.

To understand the transient splitting, we perform exact diagonalization calculations[24,25] for a 9-electron system at $ν$-1 (see Fig. 4a and Supplementary Note 2 for details). In the equilibrium state, electronic interactions at $ν=1$ create an exchange splitting $E_{ex}$[14] and produce a fully spin-polarized system. The top trace in Fig. 4b shows single electron injection spectrum into the $N=1$ LL for this condition. When many electrons are pumped into the $N=1$ LL, a fraction of them relax, due to the phonon-induced spin-flip processes[26–28], to the $N=0$ LL prior to the application of the probe pulse and create spin-down minorities in the $N=0$ LL. The energy-angular momentum diagram in Fig. 4a displays the possible spin configurations that arise when a spin minority electron is introduced into a spin-polarized system containing 8 electrons. First, a high energy state (blue) corresponds to a single minority spin immersed in a fully spin-polarized state, denoted as $ν=1+$. Second, the global ground-state corresponds to a spin-depolarized skyrmion state (black). However, as the transition

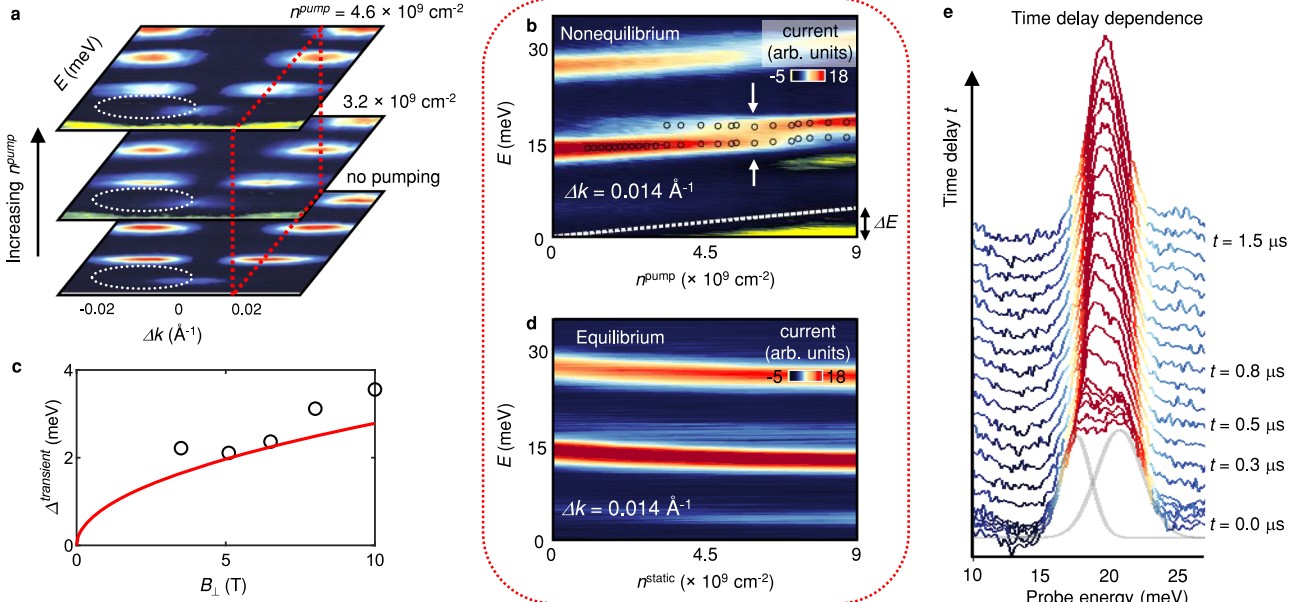

**Fig. 3 | Transient tunneling anomaly. a** Tr-MERTS spectra measured at $\nu = 1$ for different $n^{pump}$. The spectra are taken immediately after pumping the $N = 1$ LL at $t = 0$ s. Double-peak structure develops with increasing $n^{pump}$ (see the dashed circles). **b** A $\Delta k = 0.014$ Å$^{-1}$ cut of the spectrum measured as a function of $n^{pump}$ at $t = 0$ s. Open circles mark the tunneling peaks determined from Gaussian curve fitting. **c** $B_\perp$ dependence of the transient level splitting $\Delta^{transient}$. The open circles denote the measured splitting. The red curve is the splitting calculated from the exact diagonalization model: $\Delta^{transient} = 0.29 \times E_{ex}$, where $E_{ex} \approx 0.88$ meV at $B_\perp = 1$ Tesla, and $E_{ex}$ increases with the square root of $B_\perp$[15]. (see Fig. 4b). **d** A $\Delta k$ cut of the spectrum measured as a function of static electron density $\Delta n^{static}$ referenced to $\nu = 1$. Notice that the equilibrium system does not exhibit the splitting displayed in (**b**) (see also Supplementary Fig. 9). **e** Waterfall plot demonstrating time delay dependence of a splitting at $B_\perp = 8$ T and $n^{pump} = 5.5 \times 10^9$ cm$^{-2}$. The grey lines represent Gaussian curves that depict double-peak structure observed at $t = 0$. The split levels merge into a single peak after ~500 ns.

between $\nu = 1+$ and skyrmions requires energy and spin dissipation, the skyrmion state is unlikely to form immediately after the addition of extra electrons. Instead, the system starts with a nonequilibrium population of several of the excited states, including a substantial contribution from the $\nu = 1+$ state with single flipped spins. The subsequent spin-up electrons injected into the $N = 1$ LL probe this new nonequilibrium state, which is manifested by the appearance of the additional peak at energy higher than that of the first injected electrons (bottom trace in Fig. 4b). The double-peak structure observed in Fig. 4b qualitatively agrees with this expectation; the low-energy peak at $0.6E_{ex}$ corresponds to the ferromagnetic $\nu = 1$ state (red trace) while the higher-energy peak at $0.8E_{ex}$ arises from the transient $\nu = 1+$ state (blue trace). In this model, the gradual transition between the $\nu = 1$ and $\nu = 1+$ states in Fig. 3b suggests the coexistence of localized $\nu = 1$ and $\nu = 1+$ domains. Furthermore, supporting this model, Fig. 3c shows that the measured splitting between the two tunneling peaks is comparable to the calculated splitting between the $\nu = 1$ and $\nu = 1+$ states.

Finally, the transient double-peak injection spectrum emerges at relatively small $n^{pump}$ and reverts to the single-peak form as the system re-equilibrates (see Fig. 3e). Within our model discussed above, the relaxation process would involve smooth rearrangements of the local spins in the $\nu = 1+$ droplet and a transition into the skyrmion state[17,21,23] (see the graphical representation of the model in Fig. 4c). However, the critical $n^{pump} \approx 5 \times 10^9$ cm$^{-2}$, corresponding to an average spacing between injected electrons of $\approx 140$ nm, at which the $\nu = 1+$ state grows to dominate the spectrum and the relaxation mechanisms remain to be understood.

In conclusion, Tr-MERTS provides access to the nonequilibrium phenomena in a 2DES. The spectra that demonstrate the temporal dynamics of a 2DES in a wide range of tunable parameters such as $\nu$, $T$, and $B_\perp$. Furthermore, the $n^{pump}$ dependence of Tr-MERTS spectra reveals an unexpected nonequilibrium spin state occurring at ultralow temperatures. These results demonstrate the potential broad

applicability of Tr-MERTS for exploring nonequilibrium physics in LLs and other correlated electronic systems realized in 2D materials[29–33].

## Methods
### Floating-gate tunnel device
We employed a floating-gate GaAs bilayer device[17] comprising of high-mobility 180 Å and 280 Å wide GaAs quantum wells grown epaxially. The quantum wells are separated by a 130 Å wide Al$_{0.25}$Ga$_{0.75}$As tunnel barrier. The floating-gate method allow us to tune the electron density in each quantum well without contacts to the quantum wells.

### Pump-probe pulsed tunneling measurement setup
In Tr-MERTS measurements, we apply multiple voltage pulses to the tunnel device (see Supplementary Fig. 4) as described in the main text. First, we apply a pump pulse with a width ranging from 0 to ~1 μs, then apply a delay pulse with a width ranging from 0 to ~50 μs, and finally apply a probe pulse with a width of 180 ns. After applying a sequence of three pulses, we wait ~1 ms for the pumped electrons to fully tunnel back to the source before applying the next sequence of pulses. To measure the tunneling current[14,16], we record the image charge of tunneled electrons as a function of time using high electron mobility transistors. We determine the tunneling current from the initial slope of a $RC$ charging curve in the recorded time trace, where $R$ and $C$ are the tunneling resistance and the capacitance of a tunnel junction.

In order to tune the momentum of tunneling electrons, we apply $B_\parallel$ that shifts the momentum $\Delta k = eB_\parallel d/\hbar$ of the tunneling electrons[16], where the distance between electrons in the target and the source layers is 31 nm. To measure the Tr-MERTS spectra, we applied $B_\parallel$ ranging from 0 T to 6.5 T while keeping $B_\perp$ constant.

### Calibration of pumping electron density
When electrons are pumped into the target, a charging effect gives rise to an energy shift $\Delta E$ of the Fermi level of the target relative to the

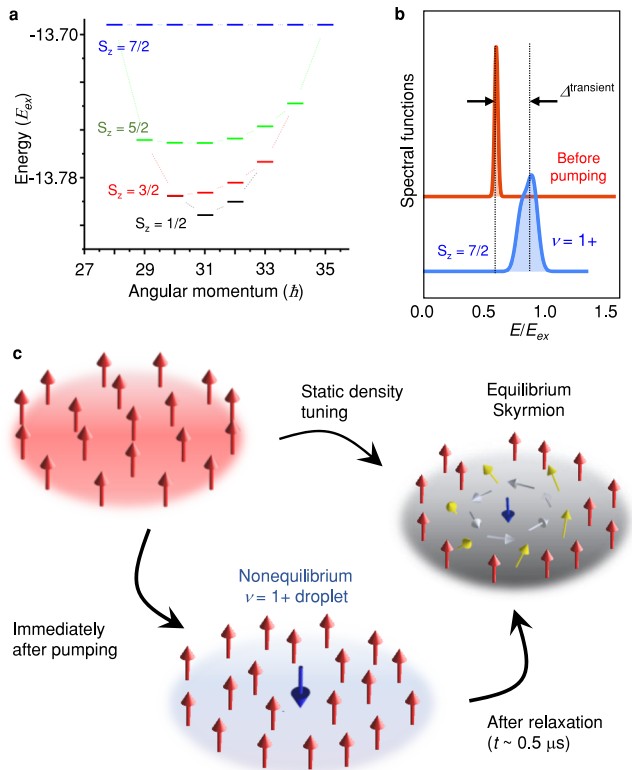

**Fig. 4 | Exact diagonalization calculation and model for the transient behavior observed at $\nu = 1$. a** Eigenenergy spectrum of the lowest Landau level with 9 electrons occupying 8 spin degenerate states. Note that the $\nu = 1$ state would be comprised of 8 spin-polarized electrons. Blue dashes depict a system with 8 spin polarized electrons and 1 spin-flipped electron, referred to here as the $\nu = 1+$ state. The black dash at angular momentum 31 depicts the ground state comprised of a maximally depolarized skyrmion. **b** Calculated spectral functions for injecting a single, spin-up, electron into the $N = 1$ Landau level probing the $\nu = 1$ (red line) and $\nu = 1+$ (blue line) states. The horizontal axis is the injection energy in the $N = 1$ LL in units of $E_{ex}$. The two dashed vertical lines depict the double-peak structure when the domains of $\nu = 1$ and $\nu = 1+$ coexist in nonequilibrium (see Supplementary Note 2 for details). **c** Graphical description of the model of the transient behavior observed at $\nu$ close to 1. Before pumping the target at $\nu = 1$, exchange interactions[14] favor the ferromagnetic state at $\nu = 1$. Shortly after pumping, the formation of $\nu = 1+$ droplets with single-flipped spins gives rise to an additional tunneling peak at higher energy. As electrons in the system rethermalize and form an equilibrium skyrmion phase[17,21], the double-peak structure evolves into the single-peak form, consistent with the equilibrium tunneling spectrum[14,17] measured by static density tuning.

Fermi level of the source. Since the charging effect is proportional to the pumped electron density $n^{pump}$, we determine $n^{pump}$ using a following algebraic expression:

$$n^{pump} = \frac{Q^{pump}}{eA} = \frac{C}{e^2 A}\Delta E = \frac{\varepsilon}{e^2 d}\Delta E, \qquad (1)$$

where $C$ and $A$ are the capacitance and area of a tunnel junction, respectively. The physical distance $d$ between the source and target is approximately equal to 36 nm. From the above equation, we calibrate $n^{pump}$, which ranges between 0 and $1 \times 10^{10}$ cm$^{-2}$.

## Data availability

The source data generated in this study have been deposited in Figshare repository https://figshare.com/s/f56e4bf9b080c31a79b8. All other data that support the plots within this paper and other findings of this study are available from the corresponding author upon request.

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

## Acknowledgements
We thank L. Levitov for a helpful discussion. This work is supported by the Basic Energy Sciences Program of the Office of Science of the U.S. Department of Energy through contract no. FG02-08ER46514. The work at Princeton University was funded by the Gordon and Betty Moore Foundation through the EPiQS (Emergent Phenomena in Quantum Systems) initiative grant GBMF4420 and by the National Science Foundation MRSEC (Materials Research Science and Engineering Centers) grant DMR-1420541. The work at Ottawa University was funded by Natural Sciences and Engineering Research Council of Canada Quantum Circuits in 2D Materials Strategic grant STPG-521420 and Natural Sciences and Engineering Research Council of Canada grant RGPIN 2019-05714 and NRC QSP-078 project.

## Author contributions
H.M.Y. and R.C.A. conceived the experiments. L.P., K.W.B. and K.W. contributed in the epitaxial growth of the sample. H.M.Y. carried out the measurements. H.M.Y. and R.C.A. analyzed the experimental data. M.K., D.M., and P.H. performed the theoretical analysis. H.M.Y., M.K., P.H., and R.C.A. wrote the manuscript with input from all authors.

## Competing interests
The authors declare no competing interests.
