## [Peer Review File · Nature Communications]

REVIEWER COMMENTS

Reviewer #1 (Remarks to the Author):

The manuscript by H. M. Yoo et al. presents a study of 2D electron systems, which are stabilized in a planar tunnel junction that permits momentum- and energy-resolved measurements of the Landau level dispersion imposed under strong magnetic fields and low (mK) temperatures. This fascinating methodology was presented by the group previously in *Science* 358, 901-906 (2017) as a new way to access very low energy and momentum scales in sample configurations that are not possible to achieve in conventional angle-resolved photoemission spectroscopy experiments. Here, the authors further extend the approach to a pump-probe scenario with temporal resolution. By electrically exciting carriers from one 2DEG into the other, they are able to observe relaxation dynamics of electrons between the two lowest Landau levels and can disentangle the spin configuration of the electrons based on the suppression of relaxation time. The authors demonstrate fine-control of pump pulse width and excited carrier density, which lead to the observation of a density-dependent splitting of Landau levels interpreted as a novel non-equilibrium spin state. I have no reservations recommending this work for publication in *Nature Communications*, as it presents exciting physics using a novel approach.

A few suggestions for minor revision:

- The abstract and introduction should mention more clearly what material system is selected to host the 2DEGS and why these are selected. At times, it can be a bit difficult to follow the details of the experimental setup.
- The time-resolution should be clearly pointed out and contrasted with pump-probe experiments using lasers. While I agree with the drawbacks of the laser-based methods mentioned by the authors, there are also many advantages such as ultrafast time-resolution and access to study the light-matter interaction. As the authors are introducing a new method in the vein of such experiments, they should present a more nuanced comparison, also explaining some of their limitations (and further opportunities).
- The drawn conclusions are extended to 2D materials in general. How system specific are the results presented here and what would be required to use this method to 2D materials in general? In other words, how applicable is the method when the electronic states close to EF exist at large momentum as in graphene and many TMDs – and how possible is it to tunnel between the 2DEGS in those materials?

Reviewer #2 (Remarks to the Author):

Review Report:

The manuscript by Yoo et al. has shown an interesting experimental technique to study the many-body interactions in time, momentum and energy spaces in the strongly correlated materials. This work surely will be an important addition in studying the physics associated to the strongly correlated materials. However, due to the lack of experimental clarities, difficulties as a reader's perspective and some other scientific doubts, I cannot recommend to publish this work in its present form. An updated version of the manuscript with clearly answering all the comments will be appropriate to publish in the *Nature Communications*. Please find all my comments as follows:

1. Critical experimental parameters such as the energy resolutions and pulse widths of the pump, delay and probe lasers are missing to show the data reproducibility of the system.
2. Why do you use such large Al_{0.25}Ga_{0.75}As tunnel barrier (130 Å)? Electron tunneling exponentially decays with the distance. So, why not using a shorter tunneling barrier of just a few

nm? It'd be important to see the results with a shorter tunneling barrier.

3. On page 3, line # 59 you have mentioned, "The first "pump" pulse ejects electrons from one of the two layers (source) and pumps these electrons into the other layer (target)."

How do you control that electron will emit from only one layer? A detailed discussion on this is necessary.

4. On page 3, line # 71 you have described the change of momentum by applying an in-plane magnetic field $B_{||}$. However, throughout the entire manuscript you only mention about applying a perpendicular magnetic field B_{\perp} . Why is that?

5. On page 4, line # 83: "When the pumped energy level in the target matches available energy states in the source, the pumped electrons tunnel back to the source..."

At what timescale this transition (reverse tunneling) happens? Could you please provide some experimental results and/or theoretical modeling?

6. On page 6 line # 121 you have mentioned that "...we reduce the width of the pump pulse and fine-tune n_{pump} injected into the target."

Again, in addition to the comment 1 please mention the reduced pulse width.

7. Nothing was mentioned about the Fig. 4d.

General comments:

1. For all the figures, sub-sectioning should be in a uniform systematic order.

2. In introduction, the discussion suddenly starts from the Fig. 1c (line # 53). It looks odd to start from Fig. 1c without mentioning the Fig. 1a-b.

3. Although, you have mentioned the name of the materials used in this work in the Methods section. It's good to mention the name of the materials at least once in the main manuscript.

4. Sometimes as a reader it's difficult to follow the story due to the disorganized discussion of the corresponding figure sub-sections. For example, discussion about the Fig. 4b came before the Fig. 4a (line # 138) and Fig. 4c even before the Fig. 4b (line # 128).

Reviewer #3 (Remarks to the Author):

The manuscript by H.M. Yoo and co authors describe a novel approach to study highly correlated states in 2D electron gas system. The experimental approach, time, momentum and energy resolved pump probe tunneling spectroscopy is a innovative way to investigate the relaxation processes in highly correlated systems. The main experimental results presented in the manuscript are unexpected splitting in the nonequilibrium energy spectrum near the ferromagnetic state, which is further explained by theoretical model as maximally spin-polarized higher energy state. Such a model provides new understanding about physics in the vicinity of filling factor in the quantum hall effect, which has been explained by skyrmion formation so far. Moreover, the time evolution of the splitting is attributed to single-flipped spins forming topological spin textures. The manuscript is written in a very comprehensive way, and the conclusions are very convincingly supported by experimental results and theoretical calculations. The manuscript bring a new understanding of highly correlated ferromagnetic states. I recommend the manuscript for publication in the present form.

Reviewer #1 (Remarks to the Author):

The manuscript by H. M. Yoo et al. presents a study of 2D electron systems, which are stabilized in a planar tunnel junction that permits momentum- and energy-resolved measurements of the Landau level dispersion imposed under strong magnetic fields and low (mK) temperatures. This fascinating methodology was presented by the group previously in Science 358, 901-906 (2017) as a new way to access very low energy and momentum scales in sample configurations that are not possible to achieve in conventional angle-resolved photoemission spectroscopy experiments. Here, the authors further extend the approach to a pump-probe scenario with temporal resolution. By electrically exciting carriers from one 2DEG into the other, they are able to observe relaxation dynamics of electrons between the two lowest Landau levels and can disentangle the spin configuration of the electrons based on the suppression of relaxation time. The authors demonstrate fine-control of pump pulse width and excited carrier density, which lead to the observation of a density-dependent splitting of Landau levels interpreted as a novel non-equilibrium spin state. I have no reservations recommending this work for publication in Nature Communications, as it presents exciting physics using a novel approach.

We thank reviewer #1's positive comments. We have carefully reviewed the referee's comments and reflected them in the revised manuscript.

A few suggestions for minor revision:

- The abstract and introduction should mention more clearly what material system is selected to host the 2DEGS and why these are selected. At times, it can be a bit difficult to follow the details of the experimental setup.

We thank reviewer #1's suggestions on the abstract and introduction. As reviewer #1 suggested, we mentioned that we used a high-mobility GaAs two-dimensional electron gas for demonstrating our technique as follows.

Revision in the abstract

Original: Using Tr-MERTS, we visualize electronic decay processes in Landau levels with lifetimes up to tens of microseconds.

Revised: Using Tr-MERTS, we visualize electronic decay processes in Landau levels formed in a high-mobility GaAs 2DES with lifetimes up to tens of microseconds.

Revision in the introduction

Original: In this study, we demonstrate the capability of Tr-MERTS via its application to interacting electrons in Landau levels (LLs) and explore the spin-dependent temporal dynamics.

Revised: In this study, we demonstrate the capability of Tr-MERTS via its application to interacting electrons in Landau levels (LLs) formed in a GaAs 2DES. We employed GaAs 2DES due to its high-mobility and ease of sample fabrication, making it an ideal prototypical 2DES for demonstrating our pump-probe method. Using Tr-MERTS, we explore the spin-dependent temporal dynamics in a LL.

- The time-resolution should be clearly pointed out and contrasted with pump-probe experiments using lasers. While I agree with the drawbacks of the laser-based methods mentioned by the authors, there are also many advantages such as ultrafast time-resolution and access to study the light-matter interaction. As the authors are introducing a new method in the vein of such experiments, they should present a more nuanced comparison, also explaining some of their limitations (and further opportunities).

We thank reviewer #1's comments on comparison between ultrafast optical pump-probe spectroscopy and electrical pump-probe spectroscopy presented in our manuscript. As reviewer #1 has pointed out, the laser-based pump-probe methods have advantages, including picosecond time-resolution. In the introduction, we have now clarified that the laser-based method has an advantage in studying ultrafast dynamics of bulk materials. We have also explained that the electrical pump-probe method allows for the study of ultra-low temperatures and slower dynamics but with a high energy resolution of submillivolts. Lastly, we note that the time resolution of our pump-probe tunneling method is limited by the RC time of a tunnel junction (the rate at which electrons are tunneled in) and can be potentially increased upto a subnanosecond with a narrower tunnel barrier.

Revision in introduction

Original: However, despite this promise, there are severe limitations in applying standard optical pump-probe spectroscopy^{3,12} to low-dimensional correlated materials.

Revised: While prior optical pump-probe studies^{3,12} have demonstrated the capability of probing dynamics occurring in correlated materials on a picosecond to subpicosecond timescale, there are limitations in applying this laser-based technique to a two-dimensional (2D) electronic system.

Original: Tr-MERTS employs short duty cycle pulses and functions in the millikelvin temperature range. In addition, fine control of electrical pulses utilized in Tr-MERTS permits high-temporal and high-energy resolution¹⁴ and precision tunability of pumping electron densities.

Revised: Tr-MERTS employs short duty cycle pulses and functions in the millikelvin temperature range. In addition, the fine control of electrical pulses utilized in Tr-MERTS permits submillivolt energy resolution¹⁴, temporal resolution of $\Delta t \lesssim 100$ ns, and precise tunability of pumping electron densities.

- The drawn conclusions are extended to 2D materials in general. How system specific are the results presented here and what would be required to use this method to 2D materials in general? In other

words, how applicable is the method when the electronic states close to E_F exist at large momentum as in graphene and many TMDs – and how possible is it to tunnel between the 2DEGS in those materials?

We thank reviewer #1 for thoughtful questions. As reviewer#1 has pointed out, in 2D materials, the electronic states at E_F are typically at large momentum. However, previous experimental studies have demonstrated that tunneling between these two-dimensional materials can still be achieved using a graphene-graphene tunnel junction [L. Britnell et al Nature Communications **4** (2013), A. Mishchenko et al Nature Nanotechnology **9** (2014), M.T. Greenaway et al Nature Physics **11** (2015), and K. Lin et al PRL **129** (2022)] and a TMD-TMD tunnel junction [K. Kim et al Nano Lett. **18** (2018)], as the electronic states in the two layers are not far apart from each other in momentum space. Therefore, our pump-probe tunneling method can be applied to these tunnel devices made of two-dimensional materials. We have now included references to these works in the conclusion.

Reviewer #2 (Remarks to the Author):

Review Report:

The manuscript by Yoo et al. has shown an interesting experimental technique to study the many-body interactions in time, momentum and energy spaces in the strongly correlated materials. This work surely will be an important addition in studying the physics associated to the strongly correlated materials. However, due to the lack of experimental clarities, difficulties as a reader's perspective and some other scientific doubts, I cannot recommend to publish this work in its present form. An updated version of the manuscript with clearly answering all the comments will be appropriate to publish in the Nature Communications. Please find all my comments as follows:

We thank reviewer#2 for comments and feedback. We have thoroughly reviewed reviewer#2's remarks and have responded with comprehensive answers to reviewer#2's questions.

1. Critical experimental parameters such as the energy resolutions and pulse widths of the pump, delay and probe lasers are missing to show the data reproducibility of the system.

We thank reviewer#2 for the questions on the energy resolution and the widths of pump, delay, and probe pulses.

- 1) The energy resolution of tunneling spectrum can be determined from the tunneling linewidth at the resonance condition due to disorder broadening [Eisenstein et al, Solid State Communications **143**, 365 (2007)]. While the disorder broadening in our device is approximated to be ~ 0.1 meV, the energy resolution in an ideal 2DES without disorder should be given by $\hbar/\tau_{\text{single-particle}} \sim 400$ pV. We now have mentioned the extremely high energy resolution (below millivolt) of our technique in the main text and have referenced the prior work [O. Dial et al Nature **448** (2007)].

- 2) The width of the pump pulse depends on the pumping density as explained in the manuscript. We now have included the range of the pump pulse width (0 us to ~ 1 us) in the method section.

Revision in the method section:

Added sentence: First, we apply a pump pulse with a width ranging from 0 to ~ 1 μ s, then apply a delay pulse with a width ranging from 0 to ~ 50 μ s, and finally apply a probe pulse with a width of 180 ns

- 3) As we have described in the caption of figure 1, the width of the delay pulse is equal to the time delay t . All of the pump-probe spectra except for those in fig. 3 a and b were plotted as a function of t . In figure 3b, where the spectra are taken immediately after pumping, and the time delay t is zero. We have now revised the figure 3 caption as follows:

Original: Tr-MERTS spectra measured at $\nu = 1$ for different n^{pump} . The spectra are taken immediately after pumping the $N = 1$ LL.

Revised: Tr-MERTS spectra measured at $\nu = 1$ for different n^{pump} . The spectra are taken immediately after pumping the $N = 1$ LL at $t = 0$ s.

Original: A $\Delta k = 0.014 \text{ \AA}^{-1}$ cut of the spectrum measured as a function of n^{pump} .

Revised: A $\Delta k = 0.014 \text{ \AA}^{-1}$ cut of the spectrum measured as a function of n^{pump} at $t = 0$ s.

- 4) The width of probe pulse was 180 ns. We now have included the width of probe pulse in the method section.

2. Why do you use such large Al_{0.25}Ga_{0.75}As tunnel barrier (130 \AA)? Electron tunneling exponentially decays with the distance. So, why not using a shorter tunneling barrier of just a few nm? It'd be important to see the results with a shorter tunneling barrier.

We thank reviewer#2's question. The tunneling current depends not only on the distance but also on the barrier height and the effective mass. For a high tunnel barrier such as oxide in an aluminum tunnel junction, a typical thickness is a few nanometers. In our tunnel device, we employed a thicker tunnel barrier because the barrier height is much lower and the mass (0.067 times the mass of a free electron) is much lighter than in typical metals. For instance, the resistance of a 3nm Al_{0.25}Ga_{0.75}As tunnel barrier is less than 1 ohm and will lead to an electrical short between the source and target layers. Also, we want to point out that, similar to other tunneling methods, our pump-probe technique is not limited by the barrier thickness but rather depends on the tunnel resistance.

3. On page 3, line # 59 you have mentioned, "The first "pump" pulse ejects electrons from one of the two layers (source) and pumps these electrons into the other layer (target)."

How do you control that electron will emit from only one layer? A detailed discussion on this is necessary.

We thank referee#2 for this question. In a tunnel junction, the polarity of an applied voltage typically determines the direction of electron flow. For instance, a positive voltage induces a positive tunneling current, except for the nonequilibrium case where a negative current (backflow) can occur despite a positive voltage. So, when we apply a pump voltage pulse, electrons flow in one direction (from the source to the target). The energy at which the electrons flow is determined by the height of the pump pulse as explained in the main text. Figure 1c in the original main text illustrates this pumping process. When the pump pulse is applied to the tunnel junction, we eject electrons from the source because the occupied states in the source align with the available states in the target layer. In such a case, electrons from the target cannot flow into the source because all the unoccupied states in the source are above the occupied states in the target.

4. On page 3, line # 71 you have described the change of momentum by applying an in-plane magnetic field $B_{||}$. However, throughout the entire manuscript you only mention about applying a perpendicular magnetic field B_{\perp} . Why is that?

We thank reviewer#2's comment. We tune the momentum by changing an $B_{||}$ as described in the main text. In the original manuscript, we included the momentum, proportional to $B_{||}$, in each figure and figure caption. We have now added a paragraph on the relationship between the $B_{||}$ and the momentum in the method section (please also refer to prior work reference 16).

Revision in the method section:

Added paragraph: In order to tune the momentum of tunneling electrons, we apply $B_{||}$ that shifts the momentum $\Delta k = eB_{||}d/\hbar$ of the tunneling electrons¹⁶, where the distance between electrons in the target and the source layers is 31 nm. To measure the Tr-MERTS spectra in Fig.1, 2, and 3, we applied $B_{||}$ ranging from 0 T to 6 T while keeping B_{\perp} constant.

5. On page 4, line # 83: "When the pumped energy level in the target matches available energy states in the source, the pumped electrons tunnel back to the source..."

At what timescale this transition (reverse tunneling) happens? Could you please provide some experimental results and/or theoretical modeling?

The timescale for tunneling is determined by RC time, where R and C are the resistance and capacitance of a tunnel junction. The capacitance is determined by the geometry of a tunnel junction. The tunneling resistance (or tunneling conductivity) depends on the properties of the tunnel barrier (such as height and thickness) and the densities of occupied states (i.e., pumped electron densities), as well as the density of available states (see Supplementary Material from ref 16 for details). The timescale for the

backflow current measured in our tunneling spectrum is similar to that of the forward current, on the order of ~ 100 ns.

6. On page 6 line # 121 you have mentioned that "...we reduce the width of the pump pulse and fine-tune n^{pump} injected into the target."

Again, in addition to the comment 1 please mention the reduced pulse width.

We thank referee#2 for the comment. In Figs. 1 and 2, we used the pump pulse with a width of ~ 1 μs . In Fig. 3, we vary the pump pulse width from 0 to ~ 1 μs . We now have revised the sentence as follows.

Original: To investigate the nonequilibrium dynamics in more detail, we reduce the width of the pump pulse and fine-tune n^{pump} injected into the target.

Revised: To investigate the nonequilibrium dynamics in more detail, we vary the width of the pump pulse from 0 to ~ 1 μs and fine-tune n^{pump} injected into the target.

7. Nothing was mentioned about the Fig. 4d.

We thank referee#2 for noticing this. We now have referred to Fig.4d (Fig.4c in the revised manuscript) in our maintext.

General comments:

1. For all the figures, sub-sectioning should be in a uniform systematic order.

We thank referee#2's comment. We have now rearranged the figure sectioning in a uniform and systematic order, as you suggested.

2. In introduction, the discussion suddenly starts from the Fig. 1c (line # 53). It looks odd to start from Fig. 1c without mentioning the Fig. 1a-b.

We thank referee#2's comment. We have now referred to Fig.1 a and b before mentioning Figure 1.c in the maintxt.

3. Although, you have mentioned the name of the materials used in this work in the Methods section. It's good to mention the name of the materials at least once in the main manuscript.

We thank referee#2 for the suggestion. We now have mentioned the name of the materials in the main text as follows.

Revision in introduction

Original: In this study, we demonstrate the capability of Tr-MERTS via its application to interacting electrons in Landau levels (LLs) and explore the spin-dependent temporal dynamics.

Revised: In this study, we demonstrate the capability of Tr-MERTS via its application to interacting electrons in Landau levels (LLs) formed in a GaAs 2DES. We employed GaAs 2DES due to its high-mobility and ease of sample fabrication, making it an ideal prototypical 2DES for demonstrating our pump-probe method. Using Tr-MERTS, we explore the spin-dependent temporal dynamics in a LL.

4. Sometimes as a reader it's difficult to follow the story due to the disorganized discussion of the corresponding figure sub-sections. For example, discussion about the Fig. 4b came before the Fig. 4a (line # 138) and Fig. 4c even before the Fig. 4b (line # 128).

We thank referee#2's comment. We have now rearranged the figure sectioning in Fig.4 so that it can help readers understand the discussion in the main text.

Summary of the change:

1. We replaced Fig. 3c with Fig. 4c to present the figures in a systematic order, allowing for a more coherent and organized discussion of the relevant points.
2. We moved Fig. 3c to Supplementary Information because the temperature dependence is not discussed in the main text. This change allows us to focus on the primary findings while still providing the relevant data for interested readers in the Supplementary Information.

Reviewer #3 (Remarks to the Author):

The manuscript by H.M. Yoo and co authors describe a novel approach to study highly correlated states in 2D electron gas system. The experimental approach, time, momentum and energy resolved pump probe tunneling spectroscopy is a innovative way to investigate the relaxation processes in highly correlated systems. The main experimental results presented in the manuscript are unexpected splitting in the nonequilibrium energy spectrum near the ferromagnetic state, which is further explained by theoretical model as maximally spin-polarized higher energy state. Such a model provides new understanding about physics in the vicinity of filling factor in the quantum hall effect, which has been explained by skyrmion formation so far. Moreover, the time evolution of the splitting is attributed to single-flipped spins forming topological spin textures.

The manuscript is written in a very comprehensive way, and the conclusions are very convincingly supported by experimental results and theoretical calculations. The manuscript bring a new understanding of highly correlated ferromagnetic states. I recommend the manuscript for publication in the present form.

We greatly appreciate reviewer #3's positive comments and feedback on our manuscript.

REVIEWERS' COMMENTS

Reviewer #1 (Remarks to the Author):

The authors have answered my points and those of the other reviewers to satisfaction. I am happy to recommend publication of the manuscript in its present form.

Reviewer #2 (Remarks to the Author):

The authors have successfully addressed all the comments. Thus, I recommend for the acceptance.

Reviewer #1 (Remarks to the Author):

The authors have answered my points and those of the other reviewers to satisfaction. I am happy to recommend publication of the manuscript in its present form.

We greatly appreciate reviewer #1's positive feedback on our responses to the questions and the revised manuscript.

Reviewer #2 (Remarks to the Author):

The authors have successfully addressed all the comments. Thus, I recommend for the acceptance.

We thank reviewer #2's positive feedback on our responses.